# The cross-sectional association between state-level public health funding per capita and physical health among adults in the United States

Stephen Hunter[1,2]*, Sze Y. Liu[3], Daniel M. Cook[4], Kia L. Davis[5], Brendan T. Smith[6,7], Roman Pabayo[1]

1 School of Public Health, University of Alberta, Edmonton, Alberta, Canada, 2 Women and Children's Health Research Institute, University of Alberta, Edmonton, Alberta, Canada, 3 Department of Public Health, Montclair State University, Montclair, New Jersey, United States of America, 4 School of Public Health, University of Nevada, Reno, Nevada, United States of America, 5 Division of Public Health Sciences, Washington University School of Medicine, St. Louis, Missouri, United States of America, 6 Health Promotion, Chronic Disease and Injury Prevention, Public Heath Ontario, Toronto, Ontario, Canada, 7 Dalla Lana School of Public Health, University of Toronto, Toronto, Ontario, Canada

☯ These authors contributed equally to this work.
* stephen1@ualberta.ca

## Abstract

### Objectives

This study examined the association between state-level public health funding per capita and the odds of poor physical health.

### Study design

Cross-sectional.

### Methods

Data from the 2018 Behavioral Risk Factor Surveillance System (BRFSS) were used. Participants' self-reported physical health was reported using the CDC Healthy Days Core Module. State-level public health funding per capita was obtained from the State Health Access Data Assistance Center website. Multilevel logistic regression was used to adjust for self-reported individual-level characteristics and state-level characteristics from the 2018 American Community Survey. We also tested whether household income or education attainment moderated any observed associations.

### Results

A one SD increase in state-level public health funding per capita was not associated with the odds ≥ 14 days of poor physical health (OR = 0.96, 95% CI: 0.90, 1.01). However, heterogeneity across household income was observed. Greater public health funding per capita was associated with lower predicted probabilities of reporting ≥ 14 days of poor physical health among respondents from low household income backgrounds (<$35,000

**Data availability statement:** BRFSS data is available at: https://www.cdc.gov/brfss/annual_data/annual_2018.html State Public health funding data is available at: https://statehealthcompare.shadac.org/rank/117/per-person-state-public-health-funding#2,3,4,5,6,7,8,9,10,11,12,13,14,15,16,17,18,19,20,21,22,23,24,25,26,27,28,29,30,31,32,33,34,35,36,37,38,39,40,41,42,43,44,45,46,47,48,49,50,51,52/a/7/154/false/lowest State-level confounders from the America Community Survey are available at: https://www.census.gov/programs-surveys/acs/data.html.

**Funding:** RP is Tier II Canada Research Chair in Social and Health Inequities. SH is supported by the Women and Children's Health Research Institute Postdoctoral Fellowship. This WCHRI Postdoctoral Fellowship has been funded by the Alberta Women's Health Foundation through the Women and Children's Health Research Institute.

**Competing interests:** The authors have declared that no competing interests exist.

USD) compared to participants with high household incomes (>$75,000 USD). No associations were observed among those with moderate ($35,000 – $70,000 USD) household incomes. A similar finding was observed among participants with less than high school education when compared to participants with post-secondary education.

## Conclusion

Greater state-level public health funding per capita appears to have a protective association against reporting ≥ 14 days of poor physical health in individuals with lower household incomes and may be helpful in reducing health inequities. Future research is needed to determine whether this association is causal.

## Introduction

The socioeconomic gradient in health suggests a positive relationship between socioeconomic and health [1]. It is likely that health-related quality of life (HRQoL) shows a socioeconomic gradient. Longitudinal studies have found some support regarding income and self-rated health, where greater income is associated with better self-rated health [2,3]. Self-rated general health is closely related to perceptions of physical health. Therefore, identifying factors that can reduce health inequities, and by extension physical health are important.

Public health aims to promote population health and reduce health inequities which should improve HRQoL [4,5]. To improve health and quality of life, public health units need adequate funding to retain staff and maintain programs and services [6]. Higher levels of public health spending was found to be associated with reduced infant-, cardiovascular-, diabetes-, and cancer- mortality [7] and rates of sexually transmitted diseases [8]. However, less is known about the association between public health funding and HRQoL. Therefore, the focus of this research was to examine the association between state-level public health funding and reporting of poor physical health. A secondary aim was to determine whether the association between public health spending and poor physical health was modified by household income or educational attainment of the individuals.

## Methods

The BRFSS is an ongoing federal-state partnership surveillance program that collects data on health behaviours, chronic health conditions, and use of preventive services via telephone surveys [9]. Participants consist of non-institutionalized individuals aged 18 years and above residing in the United States. Although the BRFSS is not population representative the use of survey weights helps make the data more representative to the non-institutionalized adult population. The BRFSS currently operates in all 50 states, as well as the district of Columbia [10]. For the current study, individual data from the 2018 BRFSS was linked with state level data from the State Health Access Data Assistance Center (SHADAC) and the 2018 America Community Survey. Informed written or verbal consent was not obtained by the study team as this was an analysis of publicly available secondary data (https://www.cdc.gov/brfss/annual_data/annual_2018.html). This study received approval from the Health Research Ethics Board at the University of Alberta (Pro00134036).

### Main exposure of interest

For each US state and the District of Columbia, data on per capita state public health funding for the year 2018 was obtained from the State Health Access Data Assistance Center

(SHADAC) website [11]. The public health funding variable summarizes funds allocated to infectious disease control, chronic disease prevention, injury prevention, environmental public health, maternal, child, and family health, and access to a linkage with clinical care. The Trust for America's Health created this variable based on funding information made publicly available from each state [12]. Per capita state public health funding was then z-transformed to help with interpretation.

## Main outcome of interest – physical health

Self-reported days of poor physical health was collected via one item from the Center for Disease Control and Prevention's Healthy Days Measures [13]. Specifically, participants were asked the following question to obtain data on self-rated physical health: "Now thinking about your physical health, which includes physical illness and injury, for how many days during the past 30 days was your physical health not good?" Responses could select between 0 to 30 days, none, don't know/not sure. This measure has shown good construct validity with self-perceived health status [14]. For the analysis, self-report of poor physical health in the last 30 days was binary coded as < 14 days of poor physical health and ≥ 14 days of poor physical health. This cutoff of 14 days or more was previously used by others [15–17].

## Covariates

Individual-level characteristics were included based on their strong associations with physical health. Age, sex assigned at birth (male; female); from which we interpret as gender given the focus on the manuscript is on social influences rather than biological mechanisms; racial identity, marital status, educational attainment, and annual household income. Racial identity (considered as another social construction) was coded into 6 categories: Non-Hispanic White; Non-Hispanic Black or African American; American Indian or Alaska Native/Pacific Islander; Hispanic; Asian; Other). Marital status was coded as single (divorced, widowed, separated, never married) or coupled (married, a member of an unmarried couple). Educational attainment was coded into four categories: less than high school, high school, some college, and post-secondary. Annual household income was coded as low (less than $35,000), medium ($35,000 to $75,000), and high (greater than $75,000). State-level confounders were derived from the 2018 American Community Survey (ACS) which is publicly available [18]. Specifically, we included the total population, the proportion of residents who are Black, median household income, and the proportion of the population living in poverty. Additionally, we adjusted for the type of public health structure based on previously established classifications [19]

## Statistical analysis

All analyses were performed in HLM (version 7.20) and used the final weights assigned to individuals. Multilevel logistic regression analyses were performed to account for the clustered structure of the data (individuals within states), adjust for individual- and area-level confounders, and use of survey weights. First, a null model was performed to obtain the overall predicted probability of ≥ 14 days of poor physical health and the 95% plausible value range. Next, an unadjusted model with only public health funding per capita was performed. Subsequently, an adjusted model with individual and area-level covariates included was run. Finally, cross-level interaction terms for household income and educational attainment with state-level public health funding per capita were added in separate models. Statistical significance was determined a priori (p < 0.05).

## Results

There were 348,221 participants from 50 states and the District of Columbia included in the analysis. Descriptive statistics for individual- and state-level characteristics are presented in Table 1. Approximately half (53.4%) of the weighted sample was aged 45 years or older, were females (50.0%) or were partnered (56.8%). Approximately two-thirds of the weighted sample were Non-Hispanic White (64.6%), had some college or post-secondary education (61.3%), or had medium to high incomes (64.5%). At the state-level, the median population size was 4,461,153 (IQR = 5,773,333), with 11.8% (SD = 10.7), and 12.9% (2.8), of the population on average being Black, or living in poverty, respectively. The average state public health funding

**Table 1. Descriptive characteristics of the 2018 BRFSS participants (n = 328, 221) and area level factors (n = 51) included in the analysis.**

| Individual Characteristics | Sample Frequency (Weighted Percent) | |
|---|---|---|
| *Age* | | |
| 18–24 | 18, 459 (11.02%) | |
| 25–44 | 83, 868 (35.58%) | |
| 45–64 | 130, 491 (34.08%) | |
| 65 + | 115, 403 (19.32%) | |
| *Gender* | | |
| Male | 163, 178 (49.96%) | |
| Female | 185, 043 (50.04%) | |
| *Race* | | |
| Non-Hispanic White | 269, 101 (64.59%) | |
| Non-Hispanic Black or African American | 28, 687 (11.58%) | |
| Asian | 7, 752 (5.07%) | |
| American Indian or Alaska Native/Pacific Islander | 6, 497 (1.00%) | |
| Hispanic | 25, 479 (15.75%) | |
| Other | 10, 705 (2.00%) | |
| *Marital Status* | | |
| Partnered | 196, 373 (56.81%) | |
| Single | 151, 848 (43.19%) | |
| *Household income* | | |
| Low | 122, 191 (30.12%) | |
| Medium | 104, 848 (28.18%) | |
| High | 121, 136 (36.28%) | |
| *Education* | | |
| Less than high school | 22,718 (11.84%) | |
| High school | 90, 887 (26.87%) | |
| Some college | 96, 858 (31.62%) | |
| Post-secondary | 137, 758 (29.67%) | |
| **State-level characteristics** | Mean (SD) | Range |
| *Population* | 6,405,637 (7,327,258) | 57,7601–39,500,000 |
| *Proportion Black* (%) | 11.83 (10.72) | 0.6–46.4 |
| *Median income* ($USD) | 62,034.65 (10,584.69) | 44097–85203 |
| *Proportion living in poverty* (%) | 12.91 (2.84) | 7.6–19.7 |
| *Public Health Funding* ($USD) | 42.04 (30.68) | 7.07–135.32 |

Note: IQR = Interquartile range.

per capita was $42.0 USD (SD = $30.7 USD) per capita, and the average median state income was $59,955 USD (SD = 10,585 $ USD).

The overall predicted probability for ≥ 14 days of poor physical health was 12.5% (95% plausible value range: 9.6%, 15.6%; Table 2). In the unadjusted model, a one SD increase in public health funding per capita was not associated with the likelihood of reporting poor physical health ≥ 14 days (OR = 0.97, 95% CI: 0.91, 1.04). Similar results were observed after adjusting for individual and area-level characteristics (OR = 0.96, 95% CI:0.90, 1.01; Table 2).

The household income*public health funding per capita interaction was significant (p = 0.007) for lower income households and the results are displayed in Fig 1. The results suggest that increased public health funding per capita is associated with a lower probability of reporting poor physical health > 14 days among individuals in the lowest income group and the slope is different compared to individuals from high income households. Educational attainment revealed a similar pattern (Fig 2). Participants with less than high school education had lower predicted probabilities of poor physical health compared to those with post-secondary education (Table 2).

## Discussion

Based on our cross-sectional results, the association between public health funding per capita and poor physical health was modified by individual household income and educational attainment. Specifically, more public health funding per capita was associated with a lower likelihood of reporting physically unhealthy days in participants from lowest income groups and lowest education groups, whereas the likelihood of reporting physically unhealthy days among middle- and higher-income, and higher education participants remained relatively stable and not associated across quartiles of public health funding per capita.

The non-significant association for public health funding and physical health may be explained by the cross-sectional nature of the data and the inability to detect any potential lag effects that public health funding may have on physical health [20]. The lack of association could also be explained by the concept of vertical equity, where states with greater needs, in this case worse physical health, received higher funding [21]. Given the significant interactions between household income and educational attainment with public health funding that we observed, it is possible that states with greater funding were equipped with programming, staffing, and surveillance or monitoring that enabled them address inequities in physical health by targeting lower income and lower education households [6]. Either way, it is difficult to understand the mechanism through which public health funding impacts physical health without further assessment of intermediary factors such as quality, effectiveness, or efficiency of public health processes and performance [22]. For example, prior research has found some evidence that increased state and federal funding was associated (p < 0.10) with less efficiency by local health departments. Whereas, greater efficiency was associated (p < 0.05) with a greater proportion of programs being produced internally by the health departments [23]. Future research will benefit from investigating potential lags and mechanisms linking public health funding per capita to physical health.

Our results support the vulnerable population approach described by Frohlich and Potvin [24]. Had interactions never been performed, we would have missed a key finding that public health unit funding per capita has protective associated with poor physical health for low-income and low-education participants (i.e., a vulnerable population). It is unclear whether our findings are consistent with the literature as most other studies looking at public health funding and population health did not examine whether associations were heterogenous by household income or education [20]. The lack of association between public health funding

**Table 2. Multilevel multiple logistic regression modelling the association between individual and area-level characteristics with ≥ 14 days of poor physical health using data from the 2018 BRFSS, SHADAC, and 2018 ACS among all states.**

| Fixed Effect | OR (95% CI) | AOR (95% CI)[a] | AOR (95% CI)[b] | AOR (95% CI)[c] |
|---|---|---|---|---|
| **Area-level characteristics** | | | | |
| z-PH Funding | 0.97 (0.91, 1.04) | 0.96 (0.90, 1.01) | 1.01 (0.93, 1.09) | 0.98 (0.91, 1.05) |
| *Population* | | 0.98 (0.94, 1.02) | 0.98 (0.94, 1.02) | 0.98 (0.94, 1.02) |
| *Proportion Black* | | **0.94 (0.89, 1.00)** | **0.94 (0.89, 1.00)** | **0.94 (0.89, 1.00)** |
| *Median income* | | **1.15 (1.04, 1.27)** | **1.15 (1.03, 1.27)** | **1.15 (1.03, 1.27)** |
| *Proportion living in poverty* | | **1.18 (1.07, 1.30)** | **1.18 (1.07, 1.30)** | **1.18 (1.07, 1.30)** |
| *PH Governance Typology* | | | | |
| Central | | 0.97 (0.88,1.07) | 0.97 (0.88, 1.07) | 0.97 (0.88, 1.07) |
| Mixed/Shared | | 1.08 (0.98, 1.19) | 1.08 (0.98, 1.19) | 1.08 (0.98, 1.19) |
| Decentralized | | Reference | Reference | Reference |
| **Individual-level characteristics** | | | | |
| *Age* | | | | |
| 18-24 | | **0.19 (0.16, 0.22)** | **0.19 (0.16, 0.22)** | **0.19 (0.16, 0.22)** |
| 25-44 | | **0.47 (0.44, 0.50)** | **0.47 (0.44, 0.50)** | **0.47 (0.44, 0.50)** |
| 45-64 | | **1.12 (1.06, 1.18)** | **1.12 (1.06, 1.19)** | **1.12 (1.06, 1.19)** |
| 65 + | | Reference | Reference | Reference |
| *Gender* | | | | |
| Male | | Reference | Reference | Reference |
| Female | | 1.04 (0.99, 1.09) | 1.04 (0.99, 1.09) | 1.04 (0.99, 1.09) |
| *Race* | | | | |
| Non-Hispanic White | | Reference | Reference | Reference |
| Non-Hispanic Black or African American | | **0.79 (0.73, 0.86)** | **0.79 (0.73, 0.86)** | **0.79 (0.73, 0.86)** |
| Asian | | **0.59 (0.46, 0.75)** | **0.59 (0.46, 0.75)** | **0.58 (0.46, 0.75)** |
| American Indian or Alaska Native/Pacific Islander | | **1.35 (1.16, 1.59)** | **1.35 (1.16, 1.59)** | **1.36 (1.16, 1.59)** |
| Hispanic | | **0.68 (0.61, 0.75)** | **0.68 (0.61, 0.75)** | **0.68 (0.61, 0.75)** |
| Other | | **1.25 (1.11, 1.41)** | **1.25 (1.11, 1.41)** | **1.25 (1.11, 1.41)** |
| *Marital Status* | | | | |
| Partnered | | Reference | Reference | Reference |
| Single | | **1.10 (1.04, 1.16)** | **1.10 (1.04, 1.16)** | **1.10 (1.04, 1.16)** |
| *Household income* | | | | |
| Low | | **3.73 (3.45, 4.03)** | **3.69 (3.41, 3.98)** | **3.74 (3.46, 4.04)** |
| Low*z-PH Funding | | | **0.92 (0.86, 0.99)** | |
| Medium | | **1.72 (1.59,1.86)** | **1.72 (1.60, 1.86)** | **1.73 (1.60, 1.86)** |
| Medium*z-PH Funding | | | 0.98 (0.90, 1.06) | |
| High | | Reference | Reference | Reference |
| *Education* | | | | |
| Less than high school | | **2.11 (1.92, 2.31)** | **2.11 (1.93, 2.32)** | **2.06 (1.88, 2.26)** |
| Less than high school*z-PH Funding | | | | **0.89 (0.81, 0.98)** |
| High school | | **1.59 (1.48, 1.70)** | **1.59 (1.49, 1.70)** | **1.59 (1.48, 1.70)** |
| High school*z-PH Funding | | | | 0.99 (0.92, 1.06) |
| Some college | | **1.51 (1.41, 1.61)** | **1.51 (1.41, 1.61)** | **1.51 (1.41, 1.61)** |
| Some college*z-PH Funding | | | | 1.01 (0.94, 1.08) |
| Post-secondary | | Ref | Ref | Ref |

Note: Bolded values are significant p < 0.05. OR = Odd's Ratio, 95% CI = 95% confidence intervals.

[a]Adjusted for individual and area level characteristics.

[b]Adjusted model including the interaction term (z-Public Health Funding * household income).

[c]Adjusted model including the interaction term (z-Public Health Funding * educational attainment). Abbreviations: PH = Public health

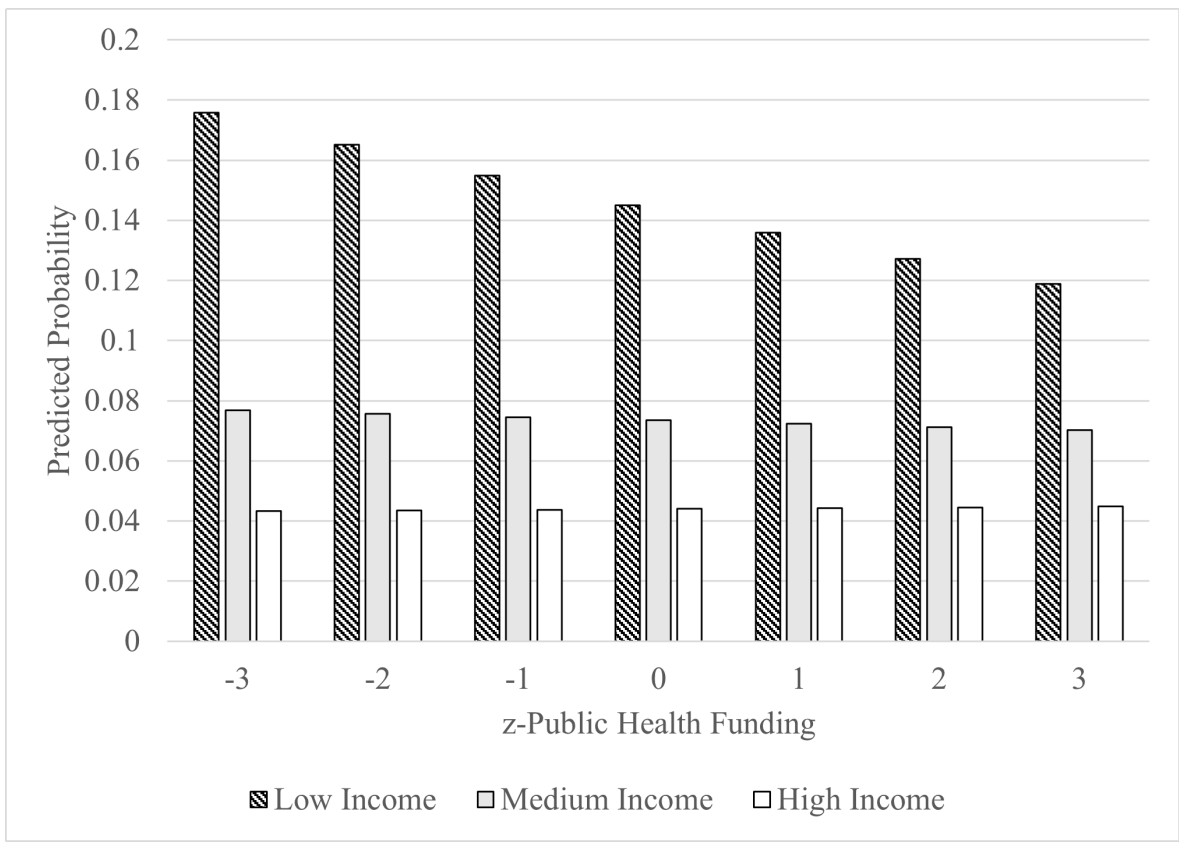

**Fig 1. Predicted probabilities of poor physical health associated with public health funding per capita by household income.**

and poor physical health among higher income and more educated participants may be explained by the fundamental causes articulated by Link and Phelan [25]. In the current study, it is possible wealthier individuals were more likely to be able to obtain resources (i.e., knowledge, money, power, prestige, social connections) to benefit their physical health and received diminishing returns from state-level public health funding. Whereas people with lower incomes or lower educations may not have these resources to improve their physical health and relied more on public services.

Strengths of this study included the representative sample collected by the BRFSS and previously validated measures of physical health [13]. There are also some limitations. For example, this study was cross-sectional making it impossible to make causal inferences (e.g., state-level public health funding per capita is causally associated with health among people with low household income). This also reduced our ability to examine any potential lag effects that might be present between public health funding and individual physical health. Another limitation is the general nature of public health funding that was used. Public health has multiple standards to deliver on, so it is difficult to know whether funding allocated to one area (e.g., disease prevention) versus another (e.g., health promotion) may improve physical health [6,11]. Future research may benefit from using more nuanced data on public health funding. Finally, while we did adjust for state-level public health structure we did not adjust for other relevant constructs resembling the organizational capacity (e.g., organizational culture, human resources) or study any intermediary factors (e.g., quality, efficiency) that may lie between public health funding and physical health [22].

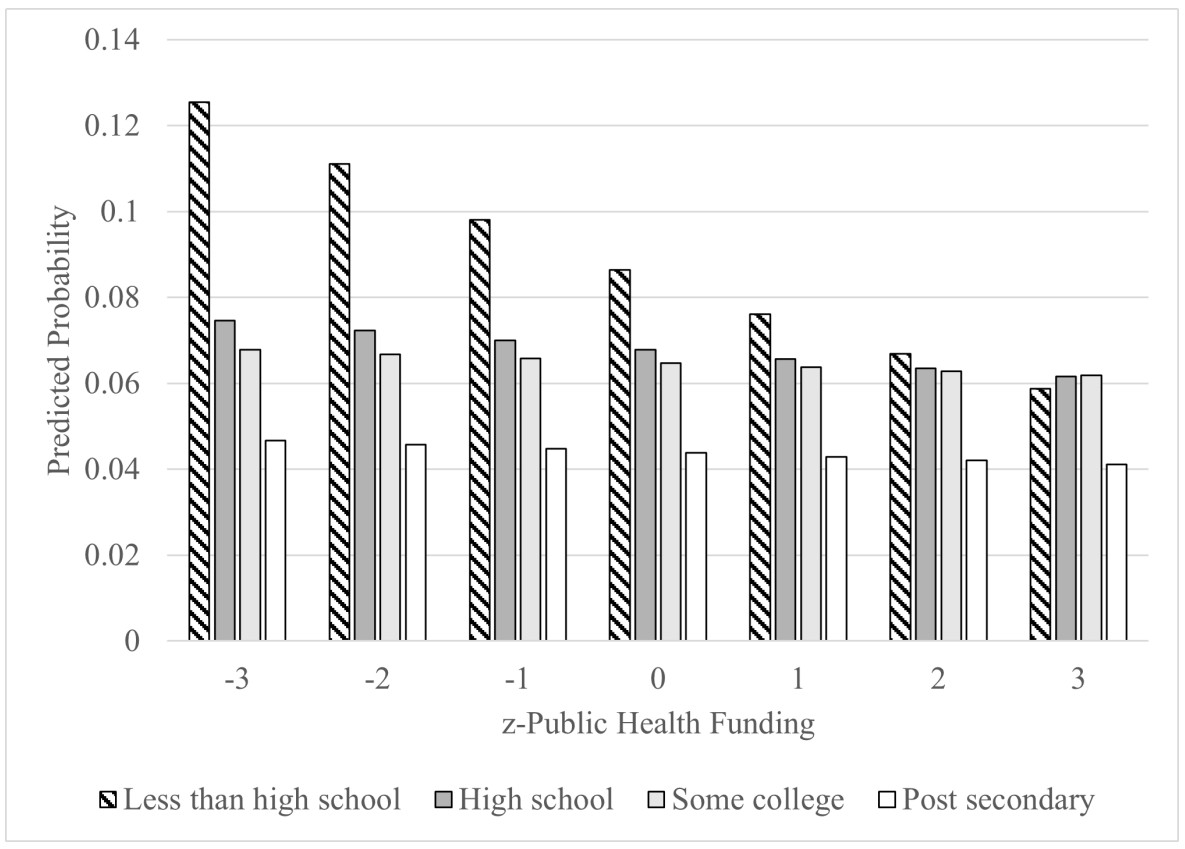

**Fig 2. Predicted probabilities of poor physical health associated with public health funding per capita by education attainment.**

## Conclusion

State-level public health funding per capita was associated with a decreased odds in reporting poor physical health among participants with lower household incomes and less than high school education. This is an important finding for public health as it suggests that states with greater public health funding at the state level may be better equipped to reduce health inequities as indicated by the social gradient [1]. Future research may benefit from using longitudinal study designs to assess potential lag effects. Further, using more specific data (e.g., fund allocation) and assessment of intermediary factors (e.g., quality, efficiency) will help clarify the association between state-level public health funding and physical health.

## Acknowledgements

None.

## Author contributions

**Conceptualization:** Kia L. Davis, Roman Pabayo.

**Formal analysis:** Roman Pabayo.

**Supervision:** Roman Pabayo.

**Writing – original draft:** Stephen Hunter.

**Writing – review & editing:** Stephen Hunter, Sze Y. Liu, Daniel M. Cook, Kia L. Davis, Brendan T. Smith, Roman Pabayo.

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
