## [Decision Letter · Decision Letter 0]

19 Nov 2024

PONE-D-24-42416The cross-sectional association between state-level public health funding per capita and physical health among adults in the United StatesPLOS ONE

Dear Dr. Hunter,

Thank you for submitting your manuscript to PLOS ONE. After careful consideration, we feel that it has merit but does not fully meet PLOS ONE’s publication criteria as it currently stands. Therefore, we invite you to submit a revised version of the manuscript that addresses the points raised during the review process.

Please address the concerns raised by the reviewers. It is important to define all the relevant variables carefully, including local level funding. The results should be interpreted carefully, and implications of the results should be incorporated. Although adding another year of BRFSS data will be useful, it is fine if you decide not to add the data set. The question about the reason for using 2018 BRFSS (rather than more recent ones) should be addressed. 

We look forward to receiving your revised manuscript.

Kind regards,

M. Mahmud Khan

Academic Editor

PLOS ONE

Journal Requirements:

2. Thank you for stating the following financial disclosure: RP is Tier II Canada Research Chair in social and health inequities. SH is supported by the Women and Children’s Health Research Institute Postdoctoral Fellowship. This WCHRI Postdoctoral Fellowship has been funded by the Alberta Women’s Health Foundation through the Women and Children’s Health Research Institute.

3. Thank you for stating the following in the Acknowledgments Section of your manuscript: RP is Tier II Canada Research Chair in social and health inequities. SH is supported by the Women and Children’s Health Research Institute Postdoctoral Fellowship. This WCHRI Postdoctoral Fellowship has been funded by the Alberta Women’s Health Foundation through the Women and Children’s Health Research Institute.

Please remove any funding-related text from the manuscript and let us know how you would like to update your Funding Statement. Currently, your Funding Statement reads as follows: RP is Tier II Canada Research Chair in social and health inequities. SH is supported by the Women and Children’s Health Research Institute Postdoctoral Fellowship. This WCHRI Postdoctoral Fellowship has been funded by the Alberta Women’s Health Foundation through the Women and Children’s Health Research Institute.

Additional Editor Comments:

We have now received comments from two reviewers. Both the reviewers have suggested significant revisions and changes, including reanalysis of data and possibly including additional years of data to strengthen the research. There appears to be significant confusions on the definitions of various important variables in the analysis. Reviewers also suggested careful editing of the paper for clarity.

Reviewers' comments:

Reviewer's Responses to Questions

**Comments to the Author**

1. Is the manuscript technically sound, and do the data support the conclusions?

Reviewer #1: Yes

Reviewer #2: No

2. Has the statistical analysis been performed appropriately and rigorously? 

Reviewer #1: Yes

Reviewer #2: Yes

3. Have the authors made all data underlying the findings in their manuscript fully available?

Reviewer #1: Yes

Reviewer #2: Yes

4. Is the manuscript presented in an intelligible fashion and written in standard English?

Reviewer #1: Yes

Reviewer #2: Yes

5. Review Comments to the Author

Reviewer #1: Review:

Line 26. “Data” is a plural noun.

Line 27. Participants’

Test for education as an effect modifier.

Report effect size for low income in abstract

Line 50, sentence beginning “Though physical” is not a sentence.

Line 55. Why just chronic disease?

First sentence beginning line 56. What are the other models? List them.

Sentence beginning line 61 is awkward.

Sentence beginning line 72 is awkward and does not follow.

This paper needs serious editing.

Line 110. Why not code days as a continuous variable? And why 14 days? How chosen?

Non-Hispanic Asian and American Indian? Why only Blacks and Whites?

Line 141. Sentence not clear.

Labeling of races in table does not follow text.

Abstract should clarify the dose response by income, not simply low and high.

Line 174, awkward.

Paragraph beginning line 213. Not relevant. The paper is not about how to raise public health funding. Delete.

Line 235. Data are available on funding for different types of health conditions.

Writing needs serious editing.

Analysis should describe effect modification by education and mention a dose response by income.

Reviewer #2: This study examined the association between state-level public health funding per capita and the odds of poor physical health (reporting more than 14 days of poor physical health) using 2018 wave of the Behavioral Risk Factor Surveillance Survey (BRFSS). The paper addresses an important public health topic but there are several important limitations.

First, the authors did not provide much justification of the use of 2018 data. It appears that 2021 data of the public health funding is also available. A repeated cross-sectional design may offer a stronger evidence and also address the limitation mentioned on page 14.

Second, the authors need to describe the key variable of interest, public health funding, in much more details. The data website suggested that this is funding from the state appropriation – would it include funding from the federal sources such as CDC, HRSA, etc.? Large county health departments may have their own funding as the paper noted with Honore et al.’s findings. The paper needs to be careful with the interpretation and the discussion of policy implications.

Third, there could be additional omitted variables that are key. States have different organizations of their public health systems, centralized, decentralized, hybrid, and mixed. The features may have impact on how public health is funded and how the funding is used.

The authors may also need to discuss the temporal aspect of the association. Will funding in 2018 have a plausible impact on self-reported healthy days surveyed across the year of 2018? It may not make sense as the impact may not have realized in a short time span.

Minor comments:

BRFSS is a federal-state partnership surveillance program not a CDC project.

HRQoL and HRQL are both used – need to be consistent.

There are multiple weights available – please specify which weight variable is used.

6. PLOS authors have the option to publish the peer review history of their article (what does this mean? ). If published, this will include your full peer review and any attached files.

**Do you want your identity to be public for this peer review?** For information about this choice, including consent withdrawal, please see our Privacy Policy .

Reviewer #1: **Yes: ** Robert A. Hahn

Reviewer #2: No

---

## [Author Response · Author response to Decision Letter 0]

3 Jan 2025

Overall: We would like to thank the Reviewer’s for taking the time to provide feedback on our manuscript. We have made changes to the manuscript in various sections and have provided excerpts and responses to each comment below. We feel that the manuscript is much stronger now based on the Reviewers’ feedback.

Reviewer #1: Review:

1) Line 26. “Data” is a plural noun.

Response: Changed “was” to “were” on line 26.

2) Line 27. Participants’

Response: Changed “Participants” to “Participants’”

3) Test for education as an effect modifier.

Response: We tested education as an effect modifier and the patterns were similar to those of household income as an effect modifier. We have included this in our statistical analysis section, results section, tables, and figures.

In the statistical analysis section on page 8 line 144:

“This was repeated for educational attainment in a separate analysis.”

In the results section on page 10, lines 170-172.

“Educational attainment revealed a similar pattern (Figure 2). Participants with less than high school education had lower predicted probabilities of poor physical health compared to those with post-secondary education. (Table 2).”

4) Report effect size for low income in abstract

Response: The effect size for the interaction with low income is not easily interpretable as it is a difference in the log odds between low income and high-income households for each SD increase in public health funding.

5) Line 50, sentence beginning “Though physical” is not a sentence.

Response: This sentence has been combined with the sentence above, and page 4 lines 51-53 now reads:

“HRQoL comprises of mental and physical components. Physical HRQoL is a stronger predictor of overall mortality, even after accounting for age, gender, and various health conditions”

6) Line 55. Why just chronic disease?

Response: Infectious disease has been added to the sentence on page 4 lines 56 and now reads:

“…understanding the burden of living with chronic or infectious diseases (7-10)”

7) First sentence beginning line 56. What are the other models? List them.

Response: The systematic review that is referenced “Bakas T, McLennon SM, Carpenter JS, Buelow JM, Otte JL, Hanna KM, et al. Systematic review of health-related quality of life models. Health and Quality of Life Outcomes. 2012;10(1):134.” highlights several models. We have included a couple in the paragraph on page 4, lines 57-61:

“Researchers have developed several models to investigate the determinants of HRQL (9). The most commonly used models include those by Wilson & Cleary (12), Word Health Organization (13), and Ferrans and Zerwic (14). Among these, the Ferrans and Zerwic (12) model is recommended because it incorporates both individual and environmental (i.e., anything outside the individual) factors (11).”

8) Sentence beginning line 61 is awkward.

Response: Added “Whereas” to the start of the sentence.

“Whereas others have…”

9) Sentence beginning line 72 is awkward and does not follow.

Response: This sentence has been restructured and now reads:

“Self-rated general health is closely related to perceptions of physical health. Therefore, it is reasonable to extend the results from these reviews to self-rated physical health (21)”

10) This paper needs serious editing.

We thank the reviewer for this suggestion. We have made significant edits throughout the paper.

For example, we have rewritten several sentences in the introduction:

Page 4 line 51-53:

“HRQoL comprises of mental and physical components. Physical HRQoL is a stronger predictor of overall mortality, even after accounting for age, gender, and various health conditions(5).”

Page 4 lines 57-61

“Researchers have developed several models to investigate the determinants of HRQoL (11). The most commonly used include those by Wilson & Cleary (12), Word Health Organization (13), and Ferrans and Zerwic (14). Among these, the Ferrans, Zerwic (14) model has been recommended because it incorporates both individual and environmental (i.e., anything outside the individual) factors (11)”

Page 5, lines 72-74

“Self-rated general health is closely related to perceptions of physical health. Therefore, it is reasonable to extend the results from these reviews to self-rated physical health (21)”

We have also combined the last two paragraphs of the introduction to enhance readability.

We have revised our results section on page 10, lines 161-165:

“The overall predicted probability for ≥ 14 days of poor physical health was 12.5% (95% plausible value range: 9.6%, 15.6%). In the unadjusted model, a one SD increase in public health funding per capita was not associated with the likelihood of reporting poor physical health ≥ 14 days (OR= 0.97, 95% CI: 0.91, 1.04; Table 2). Similar results were observed after adjusting for individual and area-level characteristics (OR =0.96, 95% CI:0.90, 1.01; Table 2).”

11) Line 110. Why not code days as a continuous variable? And why 14 days? How chosen?

Response: We have provided references for this cutoff on page 7 lines 117-118.

“This cutoff of 14 days or more has been previously used with the BRFSS healthy days module (6, 33, 34)”

12) Non-Hispanic Asian and American Indian? Why only Blacks and Whites?

Response: We are unsure what this is in reference to. Our race variable consisted of Non-Hispanic White; Non-Hispanic Black or African American; American Indian or Alaska Native/Pacific Islander; Hispanic; Asian; Other) as mentioned on page 7 lines 124-126. These terms are consistent with the terms used in the BRFSS and CDC.

13) Line 141. Sentence not clear.

Response: This sentence has been changed and now reads:

“Just over half (53.4%) of the weighted sample was aged 45 years or older, were females (50.0%) or were partnered (56.8%).”

14) Labeling of races in table does not follow text.

Response: The race categories in the Table has been changed to match the text in body.

15) Abstract should clarify the dose response by income, not simply low and high.

Response: The intent of the manuscript is to focus on state public health funding rather than individual characteristics. In the abstract we are describing the results from the interaction term, not the main effects of household income.

16) Line 174, awkward.

Response: This sentence has been changed and page 12, 191-192 now reads:

“Our study examined whether public health funding per capita was associated with physical health and explored any potential heterogeneity by household income”

17) Paragraph beginning line 213. Not relevant. The paper is not about how to raise public health funding. Delete.

Response: We agree with the Reviewer and this paragraph has been removed.

18) Line 235. Data are available on funding for different types of health conditions.

Response: This data is currently not publicly available to our knowledge. If the Reviewer could suggest where this information might be available, we would be happy to incorporate it into this manuscript if relevant. Though, if such data is available, we would likely need to run a series of models as physical health may be impacted by a number of different health conditions. Of note, by using overall public health funding we are being consistent with prior literature

19) Writing needs serious editing.

Response: The authorship team have carefully reviewed the manuscript to make any changes to enhance readability.

20) Analysis should describe effect modification by education and mention a dose response by income.

Response: We included education as an effect modifier based on the Reviewer’s comments and found a similar pattern to what we observed with household income. We suspected the similarity in patterns were due to a high proportion of low-income participants also representing participants with less than high school education. We have included these findings in the results section, tables, and figures.

In the statistical analysis section on page 8 line 144:

“This was repeated for educational attainment in a separate analysis.”

In the results section on page 10, lines 170-172

“Educational attainment revealed a similar pattern (Fig 2). Participants with less than high school education had lower predicted probabilities of poor physical health compared to those with post-secondary education (Table 2).

Reviewer #2: This study examined the association between state-level public health funding per capita and the odds of poor physical health (reporting more than 14 days of poor physical health) using 2018 wave of the Behavioral Risk Factor Surveillance Survey (BRFSS). The paper addresses an important public health topic but there are several important limitations.

1) First, the authors did not provide much justification of the use of 2018 data. It appears that 2021 data of the public health funding is also available. A repeated cross-sectional design may offer a stronger evidence and also address the limitation mentioned on page 14.

Response: We used 2018 data because it was the most complete prior to the onset of the COVID-19 pandemic. While we agree a repeated cross-sectional design could be stronger, it would also change the context around our initial research question as we would then be looking at associations between public health funding and physical health reported during the COVID-19 pandemic.

2) Second, the authors need to describe the key variable of interest, public health funding, in much more details. The data website suggested that this is funding from the state appropriation – would it include funding from the federal sources such as CDC, HRSA, etc.? Large county health departments may have their own funding as the paper noted with Honore et al.’s findings. The paper needs to be careful with the interpretation and the discussion of policy implications.

Response: We have included a description of what was captured in this variable based on what was reported by the source: Trust for America’s Health. No additional information is provided. Page 6, lines 103-107 now reads:

“The public health funding variable summarizes funds allocated to infectious disease control, chronic disease prevention, injury prevention, environmental public health, maternal, child, and family health, and access to a linkage with clinical care. The Trust for America’s Health created this variable based on funding information made publicly available from each state (31).”

We agree with the Reviewer’s second point, and based on another Reviewer’s comments, have decided to remove the paragraph where we go into more detail around policy implications and sources of funding.

3) Third, there could be additional omitted variables that are key. States have different organizations of their public health systems, centralized, decentralized, hybrid, and mixed. The features may have impact on how public health is funded and how the funding is used.

Response: We thank the Reviewers for raising this concern about differences in public health structures between states. We have classified states into decentralized, centralized, and shared/mixed based on Meit, M., Sellers, K., Kronstadt, J., Lawhorn, N., Brown, A., Liss-Levinson, R., Pearsol, J., & Jarris, P. E. (2012). Governance Typology: A Consensus Classification of State-Local Health Department Relationships. Journal of Public Health Management and Practice, 18(6), 520–528. http://www.jstor.org/stable/45038223 and included public health structure as state-level covariate in our analysis. The main findings did not change after adjusting for public health structure.

We have adjusted our tables to reflect the inclusion of PH structure as a covariate.

4) The authors may also need to discuss the temporal aspect of the association. Will funding in 2018 have a plausible impact on self-reported healthy days surveyed across the year of 2018? It may not make sense as the impact may not have realized in a short time span.

Response: We agree that there could be a lag effect that is not detectable using cross-sectional data. However, given the current manuscripts purpose was to examine whether there was an association between state-level public health funding and physical health, as well as test for heterogeneity by income, we felt that looking into potential lag effects of public health funding on physical health may be better suited for a separate manuscript where the primary focus is to determine lag effects of public health funding on physical health. We have included this

In the second paragraph of our discussion on page 12-13, lines 202-204.

“The non-significant association for public health funding and physical health may be explained by the cross-sectional nature of the data and the inability to detect any potential lag effects that public health funding may have on physical health (37)”

as a limitation on page 14, lines 237-239

“This also reduced our ability to examine any potential lag effects that might be present between public health funding and individual physical health.”

And in our concluding paragraph on page 15, lines 252-253:

“Future research may benefit from using longitudinal study designs to assess potential lag effects.”

Minor comments:

5) BRFSS is a federal-state partnership surveillance program not a CDC project.

Response: These has been changed and page 5 lines 88-90 now reads:

“The BRFSS is an ongoing federal-state partnership surveillance program that collects data on health behaviours, chronic health conditions, and use of preventive services via telephone surveys”

6) HRQoL and HRQL are both used – need to be consistent.

Response: Thank you for catching this, “HRQL” has been changed to “HRQoL” throughout.

7) There are multiple weights available – please specify which weight variable is used.

Response: We used the final weights assigned to the individuals. Land-line and cell-phone data (Raking derived weight) the variable label is _LLCPWT. We have added this information on page 8 lines 136-137.

“All analyses were performed in HLM (version 7.20) and used the final weights assigned to individuals.”

---

## [Editor Report · Decision Letter 1]

7 Jan 2025

PONE-D-24-42416R1The cross-sectional association between state-level public health funding per capita and physical health among adults in the United StatesPLOS ONE

Dear Dr. Hunter,

Thank you for submitting your manuscript to PLOS ONE. After careful consideration, we feel that it has merit but does not fully meet PLOS ONE’s publication criteria as it currently stands. Therefore, we invite you to submit a revised version of the manuscript that addresses the points raised during the review process.

 Please see the comments in the manuscript attached. Please submit your revised manuscript by Feb 21 2025 11:59PM. If you will need more time than this to complete your revisions, please reply to this message or contact the journal office at plosone@plos.org. Please include the following items when submitting your revised manuscript:

We look forward to receiving your revised manuscript.

Kind regards,

M. Mahmud Khan

Academic Editor

PLOS ONE

Journal Requirements:

Additional Editor Comments:

The paper needs major editing for readability and clarity. Initial few paragraphs are not background information and not directly relevant for the manuscript. Please edit the manuscript carefully and resubmit. I have tried to provide detailed comments on first few pages of the manuscript as guidance.

---

## [Author Response · Author response to Decision Letter 1]

28 Jan 2025

We would like to thank the Academic Editor for providing us with a chance to revise our manuscript. We found the Academic Editor’s guidance helpful and have incorporated their suggested changes in the Introduction and Methods sections of the manuscript. We have also revised the entire manuscript for clarity and feel the readability has been improved.

---

## [Editor Report · Decision Letter 2]

27 Feb 2025

The cross-sectional association between state-level public health funding per capita and physical health among adults in the United States

PONE-D-24-42416R2

Dear Dr. Hunter,

We’re pleased to inform you that your manuscript has been judged scientifically suitable for publication and will be formally accepted for publication once it meets all outstanding technical requirements.

Kind regards,

M. Mahmud Khan

Academic Editor

PLOS ONE
---

## [Editor Report · Acceptance letter]

PONE-D-24-42416R2

PLOS ONE

Dear Dr. Hunter,

I'm pleased to inform you that your manuscript has been deemed suitable for publication in PLOS ONE. Congratulations! Your manuscript is now being handed over to our production team.

Kind regards,

on behalf of

Dr. M. Mahmud Khan

Academic Editor

PLOS ONE